# Species-specific dynamics may cause deviations from general biogeographical predictions – evidence from a population genomics study of a New Guinean endemic passerine bird family (Melampittidae)

Ingo A. Müller[ORCID][1,2,3]*, Filip Thörn[ORCID][1,2,3], Samyuktha Rajan[4], Per G. P. Ericson[ORCID][2], John P. Dumbacher[ORCID][5], Gibson Maiah[6], Mozes P. K. Blom[3], Knud A. Jønsson[2], Martin Irestedt[2]

1 Department of Zoology, Division of Systematics and Evolution, Stockholm University, Stockholm, Sweden,
2 Department of Bioinformatics and Genetics, Swedish Museum of Natural History, Stockholm, Sweden,
3 Leibniz Institut für Evolutions- und Biodiversitätsforschung, Museum für Naturkunde, Berlin, Germany,
4 Department of Zoology, Division of Ethology, Stockholm University, Stockholm, Sweden, 5 Department of Ornithology and Mammalogy, California Academy of Sciences, San Francisco, CA, United States of America,
6 New Guinea Binatang Research Center, Madang, Papua New Guinea

* ingo.mueller94@gmail.com

## Abstract

The family Melampittidae is endemic to New Guinea and consists of two monotypic genera: *Melampitta lugubris* (Lesser Melampitta) and *Megalampitta gigantea* (Greater Melampitta). Both Melampitta species have scattered and disconnected distributions across New Guinea in the central mountain range and in some of the outlying ranges. While *M. lugubris* is common and found in most montane regions of the island, *M. gigantaea* is elusive and known from only six localities in isolated pockets on New Guinea with very specific habitats of limestone and sinkholes. In this project, we apply museomics to determine the population structure and demographic history of these two species. We re-sequenced the genomes of all seven known *M. gigantaea* samples housed in museum collections as well as 24 *M. lugubris* samples from across its distribution. By comparing population structure between the two species, we investigate to what extent habitat dependence, such as in *M. gigantaea*, may affect population connectivity. Phylogenetic and population genomic analyses, as well as acoustic variation revealed that *M. gigantaea* consists of a single population in contrast to *M. lugubris* that shows much stronger population structure across the island. We suggest a recent collapse of *M. gigantaea* into its fragmented habitats as an explanation to its unexpected low diversity and lack of population structure. The deep genetic divergences between the *M. lugubris* populations on the Vogelkop region, in the western central range and the eastern central range, respectively, suggests that these three populations should be elevated to full species level. This work sheds new light on the mechanisms that have shaped the intriguing distribution of the two species within this family and is a prime example of the importance of museum collections for genomic studies of poorly known and rare species.

**Data Availability Statement:** Sequences that were obtained as part of this study have been deposited at the European Nucleotide Archive (ENA) under accession numbers ERS17855383 - ERS17855413, project accession PRJEB72101. Distributional data for both Melampittidae species were extracted through the IUCN's red list webpage (https://www.iucnredlist.org/). Avian distributional data on the IUCN red list are originally provided by BirdLife (https://www.birdlife.org/). Shapefiles for administrative boundaries of Indonesia and Papua New Guinea were obtained from geoBoundaries (https://www.geoboundaries.org/). Topographic data of New Guinea was extracted from the United States Geological Survey (https://www.usgs.gov). All codes and software version numbers used for this article are mentioned within the paper and described in more detail in the supplementary material.

**Funding:** The project has been funded by the Swedish Research Council (Vetenskapsrådet, https://www.vr.se/) grant 2019-03900 (MI), the Villum Foundation, young Investigator Programme (https://veluxfoundations.dk/) project No. 15560 (KAJ), Albert & Maria Bergström Foundation 2022 (IM), Alice and Lars Siléns fund 2021+2022 (IM) and Riksmusei Vänner 2022 (http://riksmuseivanner.se/) (IM). Computations were enabled by resources provided by the National Academic Infrastructure for Supercomputing in Sweden (NAISS) and the Swedish National Infrastructure for Computing (SNIC) at UPPMAX partially funded by the Swedish Research Council through grant agreements no. 2022-06725 and no. 2018-05973. Furthermore, the authors acknowledge support from the National Genomics Infrastructure in Stockholm funded by Science for Life Laboratory, the Knut and Alice Wallenberg Foundation and the Swedish Research Council, and SNIC/Uppsala Multidisciplinary Center for Advanced Computational Science for assistance with massively parallel sequencing and access to the UPPMAX computational infrastructure. These funding sources had no role in study design, data collection and analysis, decision to publish, or preparation of the manuscript.

**Competing interests:** The authors have declared that no competing interests exist.

# Introduction

What determines the build-up of biodiversity in space and through time is a long-standing question within biology. The accumulation of phenotypic and genetic differences between populations can only be generated through reproductive isolation that impedes genetic exchange between populations [1,2]. More explicitly, gene flow between diverging populations must be sufficiently limited so that genetic exchange does not exceed the accumulation of differentiation. Barriers underlying reproductive isolation, may differ markedly. They may be postzygotic and arise from genetic incompatibilities, which produce hybrid offspring that have either reduced fitness or are infertile [3–5]. Alternatively, barriers may be prezygotic and decrease the probability of mating events between populations, due to mating preferences, or through geographical (allopatric) or habitat barriers that separate different populations [4–6].

Mountains represent a classic example of geographical barriers both as physical barriers for populations but also because they harbour highly differentiated environments at different elevations. For sedentary lowland populations, mountains may represent unsurpassable barriers, which may over time lead to isolation and differentiation of separate lowland populations. Evidence for such montane barriers restricting gene flow between lowland populations are known from various organismal groups such as amphibians, spiders and coniferous trees [7–9]. Alternatively, extensive lowland valleys can also act as barriers to geneflow between populations adapted to high elevations. Lowland environments may be unsuitable for such mountain-adapted individuals, which over time become isolated on a series of mountaintops or "sky islands" [10,11] as known from some groups of birds, lizards and plants [12–16].

Related to this is the observation that within island systems such as on New Guinea older taxa are often found at higher elevations, while young lineages that are generally widespread, good dispersers and show little differentiation inhabit the lowlands (e.g. [16–19]. Such observations (mostly from island systems) have led to the formulation of the concept of taxon cycles, in which taxa pass through phases of expansions and contractions. The concept predicts that over time, taxa move into high elevation habitats either because they are outcompeted by new young generalist taxa in their original (lowland) habitats or because they specialise and adapt to new environments at higher elevations [17,18,20–22].

Recent work on the New Guinean avifauna has provided empirical evidence in favour of species originating in the lowlands from which they move into the highlands over time and become relictual specialists [16,22,23], although some colonisation from mountaintop to mountaintop has also been shown to occur [15]. In addition, recent Pleistocene speciation events on New Guinea are mainly the result of changes in habitat distributions due to climate fluctuations, as this has caused species with continuous distributions to become geographically fragmented [24–26]. Pliocene speciation events, on the other hand, are driven mainly by geological processes such as montane uplift, which is known to have caused barriers to gene flow [27–30].

The Melampittidae represents an example of an old passerine family with only two deeply diverged species each placed in monotypic genera. Their taxonomic affinities have been difficult to establish, but recent genetic results have placed the family as sister to the clade containing crows (Corvidae), white-crowned shrikes (Eurocephalidae), crested jayshrikes (Platylophidae) and shrikes (Laniidae) with an estimated divergence time from these at ca. 16.1 Mya [31,32]. One of the species, *Melampitta lugubris* (Lesser Melampitta) is relatively common at high elevations (1150–3500 m asl.), in accordance with the notion that older species tend to occupy higher elevations [21,22,33]. The other species, *Megalampitta gigantea* (Greater Melampitta) is only known from six localities at mid-elevations (650–1400 m asl.) scattered across New Guinea. Based on few field observations, it is considered to be sedentary and to have limited flight capabilities [34,35]. Within its range, *M. gigantea* is associated with

very specific karstic habitats where it has been observed to spend considerable time nesting in narrow limestone sinkholes in which the birds have to climb in and out [34]. In contrast to *M. lugubris* the distribution of *M. gigantea* does not fit the general pattern that old taxa tend to occupy high elevation.

In this genomic study we determine the population structure and demographic patterns within the species *M. lugubris* and *M. gigantea* that each exhibit different levels of habitat connectivity. To answer this, we have re-sequenced 31 individuals at a median depth-of-coverage of 10.8 *x* and performed various population genetic (PCA, Admixture, diversity estimates) and phylogenetic analyses (mitochondrial and nuclear phylogenies). Based on the contemporary distributions of the two species we hypothesize that:

1. The individuals of *M. gigantea* represent several distinct evolutionary entities/populations, as the species is a poor disperser and has a fragmented distribution across New Guinea where it is associated with specific karst limestone habitat with sinkholes.

2. Individuals of *M. lugubris* represent a relatively cohesive group, yet with some population structure as deep lowland valleys may prevent gene flow between the various montane populations in the Central Range, the Huon mountains in the northeast and the Arfak mountains in the northwest.

## Material & methods

### DNA sampling, sequencing and read processing

In this study, we follow the taxonomy of the IOC World Bird List [36]. We sampled 24 individuals of *Melampitta lugubris* of which 22 were footpads from museum specimens and two were fresh tissue samples. Additionally, we sampled 7 individuals of *Megalampitta gigantaea*, which represent all known samples present in museum collections. One of these samples was the only available fresh blood sample for this species. The rest were footpads from historical samples (for a detailed list of samples and the museum collections in which they are stored see S1 Table). The work is mainly based on old museum specimens and therefore no ethical approval for animal research was required for this type of study. The few fresh tissue samples included in the study are from already preserved samples at natural history museums. Required permits were obtained through the Conservation and Environment Protection Authority (CEPA) of Papua New Guinea for research permits (99902749307 to K.A.J.) and export permits (017179 and 19069), which complied with all relevant regulations. The low number of modern tissue samples was due to the difficulty of acquiring fresh samples as both species are rather elusive.

DNA from fresh blood/tissue samples was extracted using Qiagen's DNeasy Blood and Tissue kits. For DNA extraction and sequencing library preparation of historical samples, we followed a modified version of Meyer and Kircher [37] that proved suitable for avian museum samples [38]. In short, we extract DNA from toepad tissue mainly following the instructions from Qiagen for animal tissue with the addition of Dithiothreitol (DTT) to improve the ligation yield. During library preparation, we treat our samples with USER enzyme to reduce deamination patterns that are typical for fragmented DNA from historical or ancient samples [39]. For a detailed protocol see [38]. Whole genome re-sequencing was performed on Illumina NovaSeq 6000 machines on S4 flow cells going through 200 cycles with a read length of 2 x 100 bp at the National Genomics Infrastructure (NGI) in Stockholm. We consider this read length adequate, as our previous work has shown an average length of historical DNA fragments around 90–130 bp [38]. Up to 24 samples with four libraries each were multiplexed on a

single flow cell lane yielding on average $10^8$ reads per sample at a targeted depth-of-coverage of 10 $x$ per individual (expected genome size: $\sim$ 1 Gb).

Sequenced reads were then polished using the reproducible *Nextflow* workflow *nf-polish* (https://github.com/MozesBlom/nf-polish) [40,41] (see S2 Table for specific github commits used). Besides providing a quality report through FastQC (v0.11.8, [42]), this pipeline performs multiple polishing steps, including deduplication (*SuperDeduper*, as part of *HTStream* v1.3.3, [43]), adapter- and quality-based trimming (*Trimmomatic* v0.39, [44]), read merging (*PEAR* v v0.9.11, [45]) and the removal of low-complexity reads, and calculates processing statistics after each step using *seqkit* (v0.16, [46]). See S2 File for the applied flags of each tool. Polished reads were then mapped onto a reference genome using *nf-μmap* (https://github.com/IngoMue/nf-umap [47]) applying *bwa-mem2* (v2.2.1, [48]) as mapping algorithm (default settings, only adding read group information), merging and converting into bam format through *samtools* (v1.13, [49]) with defaults and evaluating the quality control after mapping with *qualimap* (v2.2.2d, [50]) and investigating damage patterns that are typical for historical DNA through *DamageProfiler* (v0.4.9, [51]). We used the hooded crow (*Corvus cornix*, Refseq GCF_000738735.5, [52]) as our reference genome as it represents the most closely related species with a high-quality chromosome level assembly [31].

Our evaluation of mapped reads against the *Corvus cornix* genome showed a median depth-of-coverage (DoC) for the nuclear genome of $\sim$ 10.822 $x$ (min: 0.031 $x$, max: 31.324 $x$, SD: 7.513) and a median percentage of mapped reads at 89.5% (min 0.2%:, max: 95.8%, SD: 23.693). Detailed values for each individual and the mitochondria are listed in S1 Table.

## Phylogenetic analyses on mitochondrial and nuclear DNA

To assemble the full mitogenome from our polished reads we used *nf_mito-mania* with default settings (https://github.com/FilipThorn/nf_mito-mania) [53]. In a first step, this workflow uses a random subset of $5 * 10^6$ polished reads to assemble a de novo mitochondrial backbone using *MITObim* (v1.9.1, [54]). As a starting seed for *MITObim* we have used the *Corvus cornix* mitochondrial scaffold (Accession MT371428.1). Variant calling implemented in this pipeline filters sites with a depth-of-coverage below 20 or above three times the average depth-of-coverage across the whole mitogenome of each individual. The resulting consensus sequences of every individual were aligned using *MAFFT* (v7.407, [55], see S2 File for specific flags). An occasional artefact of *MITObim* where mitochondrial assemblies become longer than they are supposed to be resulted in overhanging sequences in some individuals. These overhangs were then cut out of the alignment after visual inspection using *Geneious Prime 2023.0.4* so that the final alignment consisted only of overlapping reads (total length 17 112 bp including gaps), which were then used as input for *RAxML-NG* (v1.1.0, [56] using the GTR+G substitution model, 100 bootstrap replicates and ten parsimony-based randomised stepwise addition starting trees to generate a mitochondrial maximum likelihood phylogenetic tree. Mitochondrial assemblies were also forced into diploid variant calls to check for contamination in our samples. As mitochondria are haploid, heterozygote sites are not expected and could therefore be indicative of cross-contamination.

For the nuclear phylogenetic tree, we used the previously mapped .bam files excluding individuals with very low mean depth-of-coverage (n = 3, DoC < 4 $x$) to call variants for each individual using *freebayes* (v1.3.1-dirty, [57]). We used *freebayes* as variant caller in this step as it is suitable to use on non-model organism genomes and is more efficient when using samples with a low to intermediate depth-of-coverage [58]. Polymorphic sites were filtered based on their quality score (> 20), allelic balance ($\geq$ 0.2), and minimum and maximum depth-of-coverage (3 $x$ / 100 $x$). We also decomposed multiple nucleotide polymorphisms (MNPs) into

single nucleotide polymorphisms (SNPs) and removed heterozygous positions and indels. These filtered .vcf files were then used as input files for the reproducible *Nextflow* workflow *nf-phylo* with default settings *(*https://github.com/MozesBlom/nf-phylo) [59]. This pipeline first creates consensus sequences for each individual and chromosome using *samtools* and *bcftools* (both v1.12, [49]) which are then combined into chromosomal alignments. Alignments were then divided into windows of different sizes (2000, 5000, 10000, 20000 base pairs, with 100 kbp between windows). Each window would undergo filtering and was included in a filtered concatenated alignment if at least half of the individuals were represented in more than 50% of an alignment column and if 80% or more of the individuals had missing data below 40%. The resulting filtered alignments were then used to generate phylogenetic trees for each window size using *IQ-TREE* (v2.0.3, [60]) while applying a GTR+I+G model. Using the same windows and model, a concatenated alignment of all autosomes and alignments for each chromosome were also generated and used to infer phylogenies with *IQ-TREE*. Additionally, summary coalescent phylogenies were generated based on all autosomal windows using *ASTRAL3* (v5.7.8, [61,62]). Lastly, site and window concordance factors (sCF and gCF) were being calculated for all inferred phylogenies using *IQ-TREE* to complement bootstrapping. All implemented flags and filters are listed in S2 File.

## Population structure and genetic diversity

To quantify population structure and to estimate genetic diversity between samples, we used a genotype likelihood approach as implemented in *ANGSD* (v0.938) as this is better suited for low coverage data and would allow us to include all of our samples [63]. Specific commands and the filters used are explained in the S2 File. The filters we used for admixture and principal component analyses (PCAs) were slightly different from those used to calculate nucleotide diversity, heterozygosity and Tajima's D. As we did not have an ancestral genome available, we used the *Corvus cornix* reference genome as ancestral sequence and folded the site frequency spectra (SFS). PCAs were performed through *PCAngsd* [64] and plotted with custom R scripts through RStudio (v 2023.03.0 build 386, R version 4.1.1, [65,66]). Admixture analyses to determine population structure were run through *NgsAdmix* [67] running up to K = 10 with ten replicates for each K and visualised with custom R scripts. Individual heterozygosity was estimated by generating a site frequency spectrum for each individual and dividing the number of sites with one derived allele by the total number of sites as performed by e.g. Hansen et al. [68]. Using SFS for each species, nucleotide diversity and Tajima's D were both estimated for each chromosome as well as in 20 kb windows sliding in steps of 10 kb using the *thetaStat* command. We divided the pairwise theta estimator (*tP*) by the total number of Sites (*nSites*) of each chromosome/window to calculate nucleotide diversity. Statistical significance of differences in heterozygosity and nucleotide diversity between the two species was checked using Welch's t-test after verifying normal distributions and inequal variances within the data. Lastly, we also calculated absolute divergences ($D_{xy}$) between different population pairs by using allele frequency estimates (maf files) from ANGSD as input. First, we calculated allele frequencies within a dataset containing all individuals to obtain a list of polymorphic sites to be analysed even if a site may be fixed in one of the populations. Next, we calculated allele frequencies for each population (see S6 Table for each included individual). Different population pairs were then compared and $D_{xy}$ was calculated using the R script calcDxy (available from https://github.com/mfumagalli/ngsPopGen/blob/master/scripts/calcDxy.R). Compared to other methods, this tool may provide underestimated values of $D_{xy}$ which should therefore only be compared between the groups within this study.

## Estimation of effective population sizes through time and divergence times

To estimate effective population sizes through time, we used Pairwise Sequentially Markovian coalescent (*PSMC*) [69] (for details on the method see S1 File). As an estimate of the neutral genomic mutation rate per generation we used $4.6*10^{-9}$ as obtained in a study of the collared flycatcher *Ficedula albicollis* [70]. We set the estimated generation time for *M. lugubris* to 3.90 years and for *M. gigantea* to 4.58 years [71]. The parameters for the *PSMC* analysis were set to "-N30 -t5 -r5 -p 4 + 30*2 + 4 + 6 + 10" following Nadachowska-Brzyska et al. [72]. The authors observed no significant change in curve shape when modifying the atomic vectors parameter (-p) and applied the same settings to several different avian species. We only ran *PSMC* for the two samples of *M. gigantea* with the highest depth-of-coverage and for each of the five identified clusters within *M. lugubris* (West, Central, East, Huon, Southeast) with 100 bootstrap replicates per individual. False negative rates (FNRs) were adjusted based on depth-of-coverage. If depth-of-coverage was higher than 15 *x*, FNR was kept at 0. However, if individual A had higher depth-of-coverage than individual B, then individual B would have an FNR of 0.1 * x, where x is the depth-of-coverage of individual A divided by depth-of-coverage of individual B.

To estimate divergence times between the two species, but also between the different sub-populations of *M. lugubris*, we first ran $F_1$-hybrid *PSMC* (*hPSMC*, [73] using the same parameters as for the previous *PSMC* analyses and implementing 100 bootstraps replicates. Additionally, we estimated mitochondrial divergences within and between the two species and subpopulations of *M. lugubris* using the previously generated mitochondrial alignment and its nucleotide diversity matrix as obtained through *Geneious Prime 2023.0.4*. We applied a simple average divergence rate between two avian lineages for the whole mitochondria at 1.8% per million years as estimated by Lerner et al. [74] which refines the "2% rule" by estimating divergence rates for several mitochondrial regions and genes rather than just cytochrome b [75]. We compared these divergence time estimates with those obtained from the *hPSMC* analyses.

## Acoustic recordings and analysis

Acoustic recordings of 10 *M. gigantea* individuals and 28 *M. lugubris* individuals were obtained from an online repository of avian vocalizations (https://xeno-canto.org/), which covered different locations across New Guinea. We included all types of vocalisations - songs, calls and vocalisations of an unknown type in the analysis, unless the function of the vocalisation was specified by the recordist (eg: alarm). This is due to the high uncertainty in estimating the type of vocalisation in *M. lugubris*, and visual comparison between vocalisations classified as 'songs' versus 'calls' between individuals recorded in the same location, often showed that they were the same. The vocalisations of each individual (median = 9 vocalisations/individual) were measured by a single author (SR) using the *Luscinia* sound analysis program (version 2.17.11.22.01, [76]. Each vocalisation was visualised using a Gaussian windowing function with the following spectrogram settings: 13 kHz maximum frequency, 5 ms frame length, 221 spectrograph points, 80% spectrograph overlap, 80 dB dynamic range, 30% dereverberation, and 50 ms of dereverberation range. Elements, which are the smallest unit of a vocalisation, were measured as continuous sound traces and then grouped into syllables within each vocalisation (each vocalisation contained only one syllable) (see S14 Fig)

The structures of the acoustic vocalisations were then compared to each other using the dynamic time warping algorithm (DTW) in Luscinia. The DTW algorithm calculates the optimal alignment between all the vocalisations i.e., syllables in the dataset, based on multiple acoustic features, and then provides a final output of a syllable dissimilarity matrix [76]. We followed the same settings used in Wheatcroft et al. [77] that has provided reliable grouping outputs for other songbird species: compression factor = 0.0001, time SD weighting = 1,

maximum warp = 25%, minimum element length = 25 samples; with the following weightings for time (5), mean frequency (1), mean frequency change (1), normalised mean frequency (1). All other acoustic features were left at the standard values. In order to examine acoustic variation in syllables, we converted the syllable dissimilarity matrix obtained from the DTW process, into 10 principal components using non-metric multidimensional scaling. These 10 components preserved the overall dissimilarity between the syllables well (kruskal stress value = 0.002, values < 0.1 are usually considered good [76]) and were used to infer acoustic variation among the two species.

## Results

During contamination control using mitochondrial assemblies, we observed an increased amount of heterozygous sites across the libraries in 6 individuals (S3 Table). Upon manual inspection using *Geneious Prime* we found that these heterozygote positions mostly appear in blocks and often within the same regions. This suggests that they were in fact nuclear mitochondrial sequences (NUMTs) that were wrongly mapped onto the mitochondrial genome instead of being a result of contamination. We also observed that non-reference alleles often appeared at a lower frequency (98.093% of heterozygote sites had a reference allele frequency > 0.5, median reference allele frequency across all heterozygote sites at 0.874) and therefore disappeared during consensus calling, as the more frequent allele gets chosen during this step. Nonetheless, we manually excluded two regions from all samples with blocks (in total 5 700 bp out of the entire alignment's 17 112 bp) of heterozygote sites shared across the majority of individuals. The remaining 11 412 bp were used to generate the mitochondrial phylogenies.

### Phylogenetic analyses on mitochondrial and nuclear DNA

We found high congruence between phylogenies built from mitochondrial and nuclear genomes (Figs 1A and S1). Different window sizes and summary coalescent vs concatenated nuclear phylogenies also had little effect on the topology. We recovered three main clusters within *M. lugubris* that correspond to the geographic location of the samples on an east to west axis (Fig 1). These clusters also align with previously described subspecies of *M. lugubris* [35]. The first cluster within *M. lugubris* consists of individuals inhabiting the Birds-Head of northwestern New Guinea as well as an individual in the westernmost part of the Central Range. The next cluster inhabits the western and central parts of the Central Range of New Guinea, and the third cluster inhabits the eastern and south-eastern section of the Central Range as well as the isolated outlying Huon mountains. *M. gigantaea*, on the other hand, shows very short branch lengths between individuals compared to *M. lugubris* indicating less diversity within this species. Relationships within *M. gigantaea* are also in accordance with the geographical locality of the samples.

### Population structure and genetic diversity

As observed in the phylogenies, we recover a similar pattern in the PCAs (Fig 2A), heterozygosity (Figs 2B and S13), nucleotide diversity ($\pi$, S4 Table and S2 Fig), absolute divergence ($D_{xy}$, S6 Table) and admixture (Fig 2C) where *M. gigantaea* exhibits lower genetic diversity in compared to *M. lugubris*. For *M. lugubris* Tajima's D was consistently negative with a mean value of -0.902 (SD: 0.126, median: -0.897, S5 Table and S3 Fig) which is indicative of either population expansion or a selective sweep. In *M. gigantaea* values for Tajima's D were slightly above zero in the range of 0–0.2 (mean 0.103, SD: 0.040, median: 0.112, S5 Table and S3 Fig). Positive values of Tajima's D could indicate a reduction in population size or balancing selection acting, however

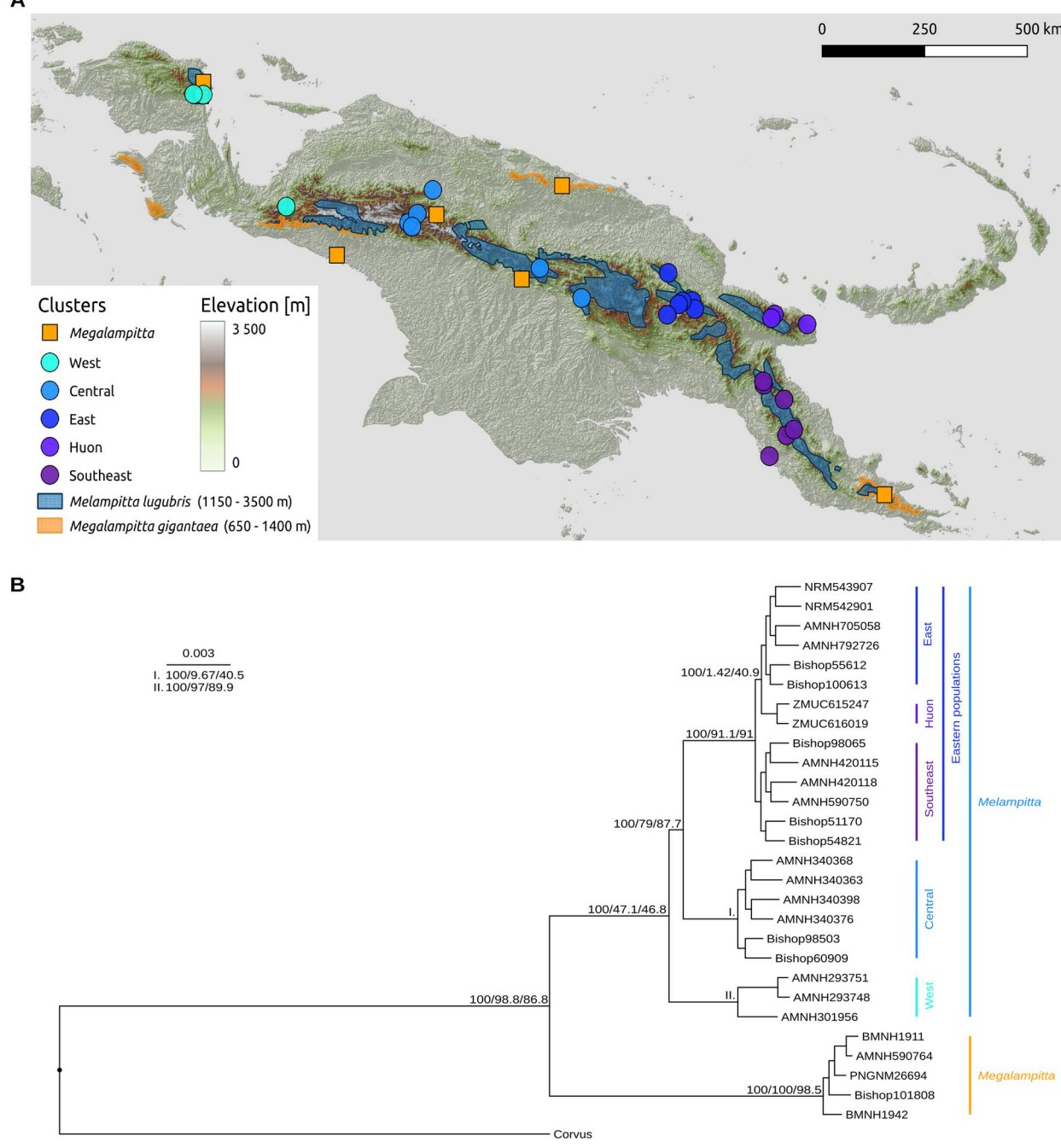

**Fig 1. A** Distribution map and sampling sites of *M. gigantaea* (orange distribution) and *M. lugubris* (blue distribution), subclusters of *M. lugubris* are also coloured differently. Shapefiles for administrative boundaries were obtained from *geoBoundaries [78]* under a CC BY license, with permission from Dan Runfola, original copyright CC BY 4.0 (2020), the map was created using *QGIS [79]*. **B** Nuclear phylogeny of Melampittidae highlighting the subdivisions within *M. lugubris* (West, Central, East, Huon, Southeast), the tree was constructed using *IQ-TREE* on concatenated 5 kbp window alignments with a GTR+I+G substitution model, support values next to the main branches show bootstraps/site concordance factors (sCF)/window concordance factors (wCF).

as the values are so close to zero the population may just evolve neutrally. In the PCA (Fig 2A), PC1 separates the two species, afterwards *M. gigantaea* remains closely clustered up to PC4, while subgroups corresponding to geographic localities make up clusters within *M. lugubris*

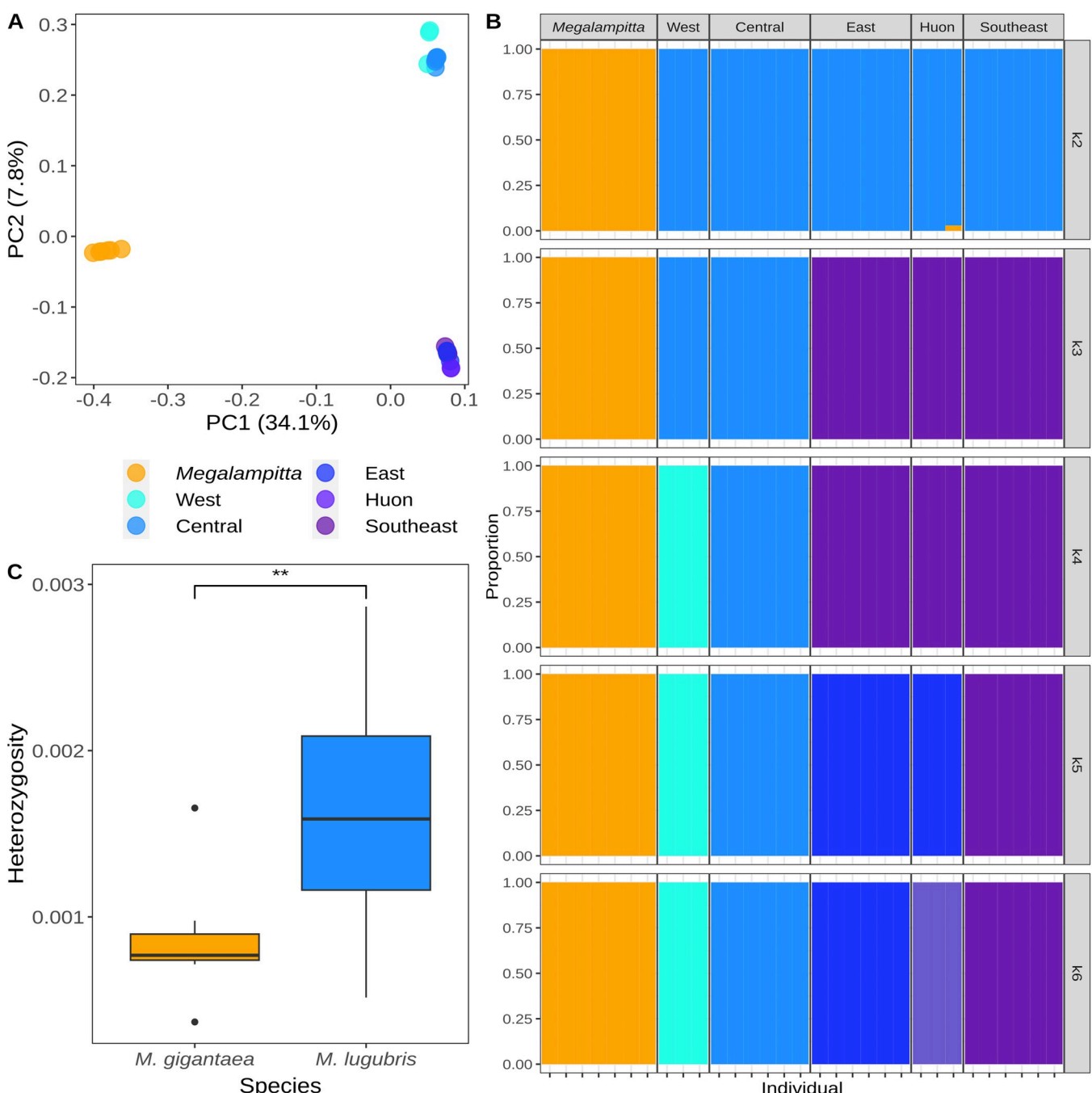

**Fig 2. Genetic diversity and population structure in Melampittidae. A)** PCA showing the first two principal components for both species (*M. gigantaea* in orange, subclusters of *M. lugubris* in shades of blue/purple). **B)** Admixture analysis from K = 2 to K = 6 **C)** Heterozygosity for all individuals between both species.

(S4 Fig). The two distinct clusters of *M. lugubris* on PC2 (Fig 2A) separate eastern New Guinean populations and northwestern New Guinean populations as also observed in the phylogenetic tree (Fig 1B). Both heterozygosity and nucleotide diversity were significantly lower in *M. gigantaea* than in *M. lugubris*. Although we observed a clear trend of increasing heterozygosity with higher depth-of-coverage, the slopes for each population were similar and consistently higher in

all but one population of *M. lugubris* (S5 Fig). Admixture analysis revealed no substructure within *M. gigantaea* from K = 2 to K = 7. For *M. lugubris* the observed clusters between K = 2–6 align with the clusters observed in the phylogenetic trees and in the PCAs. Further subdivisions within the main clusters of *M. lugubris* at higher values of K are also corresponding to the populations' geographical location. Estimates of mean absolute divergence ($D_{xy}$) per-site across the entire genome ranged from a minimum of 0.0005 between the geographically most distant *M. gigantaea* individuals to a maximum of 0.012 between the two species. $D_{xy}$ between different pairs of (sub)populations within *M. lugubris* (min: 0.002, max: 0.005) was higher than within *M. gigantaea* and about half as much as the divergence between *M. gigantaea* and *M. lugubris* in the most divergent pair of *M. lugubris* (West vs. East). An overview of all pairwise comparisons is given in S6 Table.

## Estimation of effective population size in time and divergence times

*PSMC* curves (Fig 3) for samples from the same populations had similar shapes, but not entirely overlapping as depth-of-coverage varied between samples. Within *M. lugubris* the shape of the curves varied, but most of this variation could be ascribed to population specific events. In *M. gigantaea*, we observe an effective population size peak at around 200 Kya followed by a steady decline in effective population size up until around 40 Kya.

The divergence time obtained from *hPSMCs* curves for the split between *M. gigantaea* and *M. lugubris* was estimated to about 10 mya (S6 Fig). Splits between subgroups within *M. lugubris* were estimated more recently with the split between Western+Vogelkop and Eastern populations at around 4–5 mya (S7 Fig) and between Western and Vogelkop populations at about 3–4 mya (S8 Fig). The next divisions within Eastern *M. lugubris* populations (East, Huon and Southeast) happened at similar times around 1 mya (S8 and S9 Figs).

Mitochondrial divergences were, as expected, highest for comparisons between *M. gigantaea* and *M. lugubris* and its subpopulations at a range of 9.8–13.3%. Divergence within *M. gigantaea* was also lower (mean 0.912%) than within *M. lugubris* (mean 4.979%) or even in some of its subpopulations. (For an extensive table with all comparisons of mitochondrial divergence see S11 and S12 Figs). Divergence times obtained by assuming an average rate of mitochondrial divergence of 1.8% were 5.5–7.4 mya for the split between *M. gigantaea* and *M. lugubris*, 4.1–6.2 mya for splits between Western+Vogelkop populations from Eastern populations of *M. lugubris* and Western populations from Vogelkop populations at 4–4.2 mya. Subdivisions within the Eastern populations were estimated at 0.2–2.9 mya.

## Acoustic recordings and analysis

The first ten principal components collectively explained 97% of the variation in vocalisations across the two *Melampitta* species. PC1, which explained 44.5% of the variation in all vocalisations, was more varied for *M. lugubris* (standard deviation (SD) = 0.086) compared to *M. gigantaea* (SD = 0.036). The same was true for PC2, where the standard deviation was once again higher for *M. lugubris* (SD = 0.11) compared to *M. gigantaea* (SD = 0.017). These results show that *M. lugubris* has greater acoustic diversity than *M. gigantaea* (Fig 4), highlighting the need for a more in-depth examination of life history trait variation between the two species. In addition to obtaining higher quality recordings, future work should examine whether this preliminary observation of acoustic variation corresponds to different populations.

## Discussion

The formation of the avifauna on New Guinea largely follows the predictions of taxon cycles [15,16] whereby new species form in or colonise through the lowlands and over time move

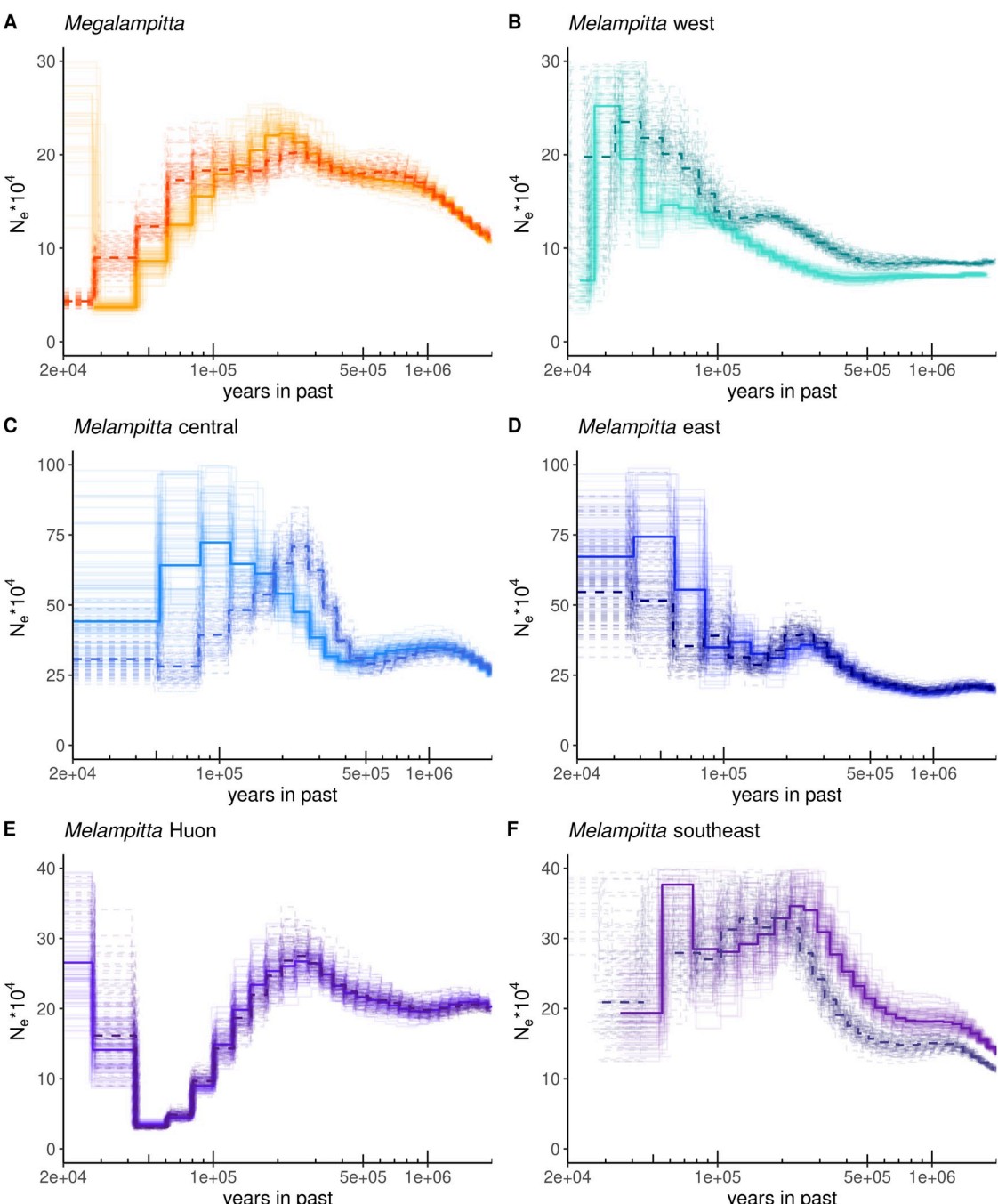

**Fig 3.** *PSMC* plots for two representative individuals of **A)** *M. gigantaea* (solid line: AMNH 590764, dashed line: PNGNM 26694) **B)** *M. Lugubris* western population (solid line: AMNH 293751, dashed line: AMNH 293748) **C)** *M. lugubris* central population (solid line: AMNH 340368, dashed line: Bishop 98503) **D)** *M. lugubris* eastern population (solid line: Bishop 100613, dashed line: Bishop 55612), **E)** *M. lugubris* Huon population (solid line: NHMD 615247, dashed line: NHMD 616019) **F)** *M. lugubris* southeastern population (solid line: Bishop 54821, dashed line: AMNH 590750) Thinner lines depict the curves of bootstrap runs.

upwards and become relictual at high elevations. The family Melampittidae is a species-poor old endemic lineage of New Guinea [31]. The family includes two extant species of which one (*Melampitta lugubris*) follows the general taxon-cycle expectation in that it is an old lineage that

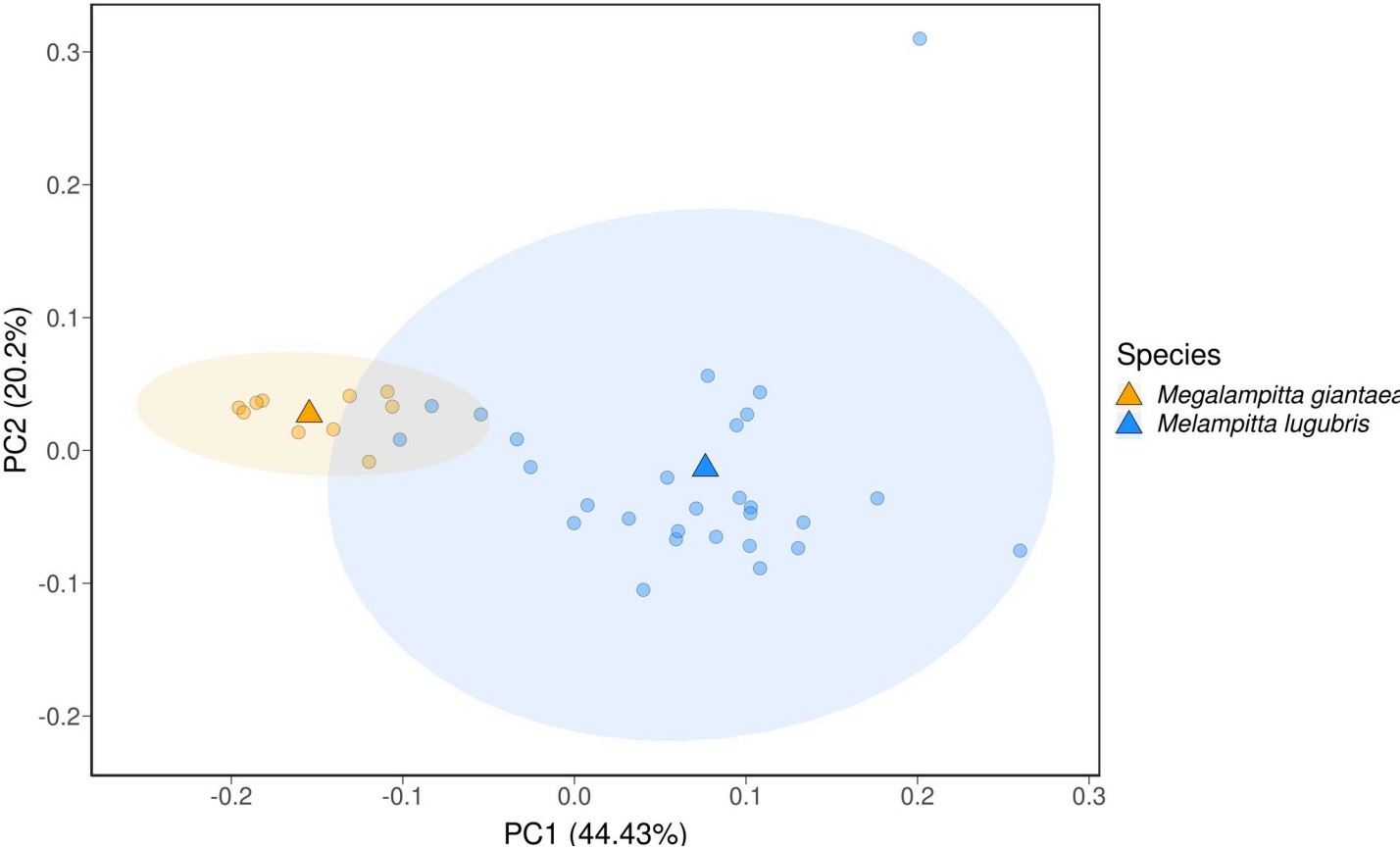

**Fig 4. Acoustic variation across species.** Principal component space (PC1-2) of vocalisations from *M. gigantaea* and *M. lugubris*. PC1 and PC2 scores are averaged within individuals and triangles represent species centroids. Ellipses contain 95% of vocalisations of each species. The significant outlier within *M. lugubris* may represent an odd vocalisation that is not directly comparable with the other vocalisations included here. Note that vocalisations for *M. gigantaea* were only available from three localities (the Fakfak mountains in the Bird's Neck, a locality in the southern Bird's Neck and Tabubil in the central highlands) and vocalisations for *M. lugubris* were only available from two of the three distinct clades (samples were available from the western and central but no vocalisation data was available from the eastern and Huon populations).

inhabits montane forests of New Guinea. The other species, *Megalampitta gigantea*, however, has a distribution associated with specific karst habitats at lower elevations and in foothills [35].

Our divergence time estimates suggest that *M. gigantea* and *M. lugubris* diverged from each other in the Miocene (at approximately 10 Mya based on hPSMC results), which is slightly younger than the divergence time estimated by Jønsson et al. [80] and slightly older than the divergence time estimated by McCullough et al. [81]. The three main populations of *M. lugubris* (Fig 1B) diverged from each other in the early Pliocene (at approximately 4–5 Mya based on hPSMC curves). A Pliocene divergence of *M. lugubris* populations coincides with major uplift of various mountain regions on New Guinea [82–84], which may have shaped the present population structure of *M. lugubris*. The distributional pattern of populations of *M. lugubris*, with one distinct Vogelkop population and a division of an eastern and a western population along the central mountain range, is also a pattern similar to that found in other New Guinean mountain birds with Pliocene divergences [30,85,86].

The PSMC curves of the three main populations of *M. lugubris* differ (Fig 3), yet with a general trend of increasing population sizes towards the present. The exception to this is the population of the Huon mountains, which shows a continuous decrease in population size since approximately 100 Kya. Our interpretation is that eastern and south-eastern populations of the

Central Range have maintained continuous gene flow, while the connectivity with the Huon population was broken or at least severely reduced as this population became isolated in the outlying Huon Mountain range.

Given a presumed poor dispersal capacity [34] and a patchy distribution at mid-elevations, we initially hypothesised that *M. gigantea* would exhibit a clear population structure. However, contrary to expectations, all our samples of *M. gigantea*, from localities scattered across New Guinea, cluster tightly together genetically (Figs 1 and 2). Analysis of vocalisations also shows a similar pattern, as *M. gigantea* exhibits less acoustic diversity compared to *M. lugubris* (Fig 4). This is fascinating and difficult to explain. Below, we discuss three scenarios that may provide possible explanations for these patterns. First, it is possible that continuous migration (or high rates of juvenile dispersal) of *M. gigantea* individuals maintains contact and gene flow between populations. However, an exclusively ground-dwelling lifestyle and the lack of long-distance flight capabilities, suggested by its morphology and field observations contradict this scenario [34,35]. Second, it is possible that their presently known fragmented distribution does not properly reflect their actual distribution, which may be more extensive [35,87]. Karst regions are generally species-poor in comparison to the species-rich tropical forests of New Guinea and such localised karstic areas dispersed throughout New Guinea may therefore have commanded less attention by ornithological surveys. Finally, it is possible that *M. gigantea* once had a wider more continuous distribution and that a recent decline has left scattered populations in small pockets of Karst habitat. The PSMC analyses support this scenario by showing that the population size of *M. gigantea* has dropped dramatically within the last 200 Ky (Fig 3). The fact that *M. gigantea* is highly adapted to a very specific habitat type (nesting in deep holes in karst limestone that they have to climb out of [33] is, however, difficult to reconcile with this scenario. However, one may speculate that *M. gigantea* in the past had broader habitat preferences not only restricted to the present karst limestone habitats. Perhaps during the last 200 Ky, increased competition from other species forced *M. gigantea* to retract to a particular low-diversity habitat type, leaving behind the scattered distribution that we see today. Overall, we find it most plausible, that *M. gigantea* had a larger and more continuous distribution in the past, yet we acknowledge that the present distribution may be underestimated. Additional ornithological surveys to suitable habitats may, thus, reveal further *M. gigantea* populations.

## Conclusions

In this study, the rather surprising population structure of the two species of an old New Guinean avian family have been elucidated by genomic data largely obtained from historical museum collections. While the population structure of *Melampitta lugubris* is similar to those found in other mountain birds of New Guinea with similar age, the population structure of *Megalampitta gigantea* is intriguing. The study is an example of how intrinsic properties, such as demographic history as exhibited by *M. gigantea*, may cause their population dynamics to deviate from general biogeographical predictions. The study is also an example of how important museum collections are for increasing the knowledge of rare taxa that occur in remote regions. The levels of divergence between the three major populations of *M. lugubris* are well above those at which ornithologists would normally assign species rank. Consequently, we tentatively propose that these three populations should be elevated to species rank, *Melampitta lugubris* (Schlegel, 1871) in the Vogelkop region, *Melampitta rostrata* (Ogilvie-Grant, 1913) in the western central range and *Melampitta longicauda* (Mayr & Gilliard, 1952) in the eastern central range.

## Supporting information

**S1 Fig. Mitochondrial phylogeny for all individuals based on an alignment of mitochondrial consensus sequences.** The tree was constructed using *RaxML-NG* applying a GTR+G substitution model.
(TIF)

**S2 Fig. Individual nucleotide diversity (π) is significantly lower in *M. gigantaea* (orange) than in *M. lugubris* (blue).** The applied statistical test was a Welch's two sample t-test for unequal variances.
(TIF)

**S3 Fig. Tajima's D across all chromosomes.** Chromosomes are divided into macrochromosomes ($> = 40$ Mbp), intermediate chromosomes ($> = 20$ Mbp, $< 40$ Mbp) and microchromosomes ($< 20$ Mbp). Values are consistently negative across most of each chromosome in *M. lugubris* (blue) and slightly positive in *M. gigantaea* (orange).
(TIF)

**S4 Fig. Principal components 1 to 4 describing divisions within subpopulations of *M. lugubris* (shades of blue/purple) while *M. gigantaea* (orange) remains a tight cluster.** Weyland represents one individual (AMNH 301956) that was collected between our Western and Central populations and is assigned to the Western population in most other analyses.
(TIF)

**S5 Fig. Heterozygosity shows a correlation with increasing depth of coverage (DoC).** Slopes are similar between populations/species and *M. gigantaea* still shows lower heterozygosity than most *M. lugubris* populations when comparing individuals with similar DoC. To fit regression lines, we applied Kendall's rank correlation coefficient as it is recommended for smaller sample sizes containing outliers *[Kendall MG. A New Measure of Rank Correlation. Biometrika. 1938;30(1/2):81–93.].*
(TIF)

**S6 Fig. *PSMC* for *M. lugubris* (blue) and *M. gigantaea* (red) and their hybrid *PSMC* curve (purple) to show the divergence time between the two species.**
(TIF)

**S7 Fig. *PSMC* for *M. lugubris* NHMD616019 from Huon (red) and *M. lugubris* B98503 from Central New Guinea (blue) and their hybrid *PSMC* curve (purple) to show the divergence time between Eastern populations and Western + Central populations of *M. lugubris*.**
(TIF)

**S8 Fig. *PSMC* for *M. lugubris* B98503 from Central New Guinea (red) and *M. lugubris* AMNH293751 from Western New Guinea (blue) and their hybrid *PSMC* curve (purple) to show the divergence time between Central populations and Western populations of *M. lugubris*.**
(TIF)

**S9 Fig. *PSMC* for *M. lugubris* NHMD616019 from Huon (red) and *M. lugubris* AMNH590750 from the Southeast (blue) and their hybrid *PSMC* curve (purple) to show the divergence time between Huon populations and Southeastern populations of *M. lugubris*.**
(TIF)

**S10 Fig.** *PSMC* **for** *M. lugubris* **NHMD616019 from Huon (red) and** *M. lugubris* **B100613 from the East (blue) and their hybrid** *PSMC* **curve (purple) to show the divergence time between Huon populations and East populations of** *M. lugubris.*
(TIF)

**S11 Fig. Mitochondrial divergence matrix showing the minimum (first value) and maximum (second value) for each comparison of populations and species,** *M. gigantaea* **(Meg),** *M. lugubris* **(Mel), Eastern populations (EPops) include the subpopulations East, Southeast and Huon.**
(TIF)

**S12 Fig. Mitochondrial divergence matrix showing the mean divergence for each comparison of populations and species,** *M. gigantaea* **(Meg),** *M. lugubris* **(Mel),** *Eastern populations* **(EPops) include the subpopulations East, Southeast and Huon.**
(TIF)

**S13 Fig. Heterozygosity between** *M. gigantaea* **and populations of** *M. lugubris. M. gigantaea* **is shown in orange, populations of** *M. lugubris* **in shades of blue/purple. Statistical significance is only shown for significant differences between pairwise comparisons of** *M. gigantaea* **and each population of** *M. lugubris* **applying Welch's t-test.**
(TIF)

**S14 Fig. Representative acoustic vocalisations of** *M. gigantaea* **and** *M. lugubris.* For each species, vocalisations from four different individuals are depicted. The top panel illustrates how vocalisations were measured in the acoustic software Luscinia: The user manually traces out the elements, the smallest unit within each vocalisation (in green), after which they are grouped into syllables (in red). Each vocalisation typically contains only 1 syllable for both species. The dynamic time warping algorithm in *Luscinia* creates a matrix of syllable dissimilarities using multiple frequency and time measurements that are extracted from these measured syllables.
(TIF)

**S1 Table. List of samples.** Additional information such as sample locality, museum voucher, tissue type, etc. are included. The table also shows mapping statistics (e.g. mapping percentage and depth-of-coverage) for each individual.
(XLSX)

**S2 Table. Used** *github* **commits when running** *Nextflow* **workflows.**
(XLSX)

**S3 Table. Filtered individuals with heterozygous blocks in mtDNA.**
(XLSX)

**S4 Table. Individual nucleotide diversity (π).** Sheet 1 (Individual) contains statistics calculated for each species using i) all chromosomes and ii) only autosomes (aut.). Sheet 2 (Species-wide) contains estimates averaged across i) all chromosomes and ii) only autosomes (aut.).
(XLSX)

**S5 Table. Species-wide Tajima's D.** Values averaged across i) all chromosomes and ii) only autosomes (aut.).
(XLSX)

**S6 Table. Mean per-site $D_{xy}$ etimates.** Sheet 1 (Dxy) shows pairwise comparisons between all major population splits as well as an estimate within *M. gigantaea*. Sheet 2 (Populations) lists

the individuals that were included in each population.
(XLSX)

**S1 File. Details on *PSMC* methodology.**
(DOCX)

**S2 File. Codes and parameters settings.**
(DOCX)

**S3 File. List of adapters that were removed during trimming.**
(DOCX)

## Acknowledgments

We thank all the staff and field assistants that facilitated fieldwork in Papua New Guinea. Notably, the Binatang Research Center and local communities in the YUS conservation area from the villages of Towet and Yawan. We are also grateful for the assistance provided by the PNG National Museum and Art Gallery and the Conservation and Environment Protection Authority (CEPA) of Papua New Guinea for research permits (99902749307 to K.A.J.) and export permits (017179 and 19069). We would like to acknowledge the following museum collections and their managers that have generously provided us with tissue samples for this study: American Museum of Natural History, New York, USA (Paul Sweet, Pete Capainolo, Tom Trombone and Brian T. Smith); Bernice Pauahi Bishop Museum, Honolulu, USA (Molly Hagemann); Natural History Museum, London, UK (Robert Prys-Jones, Hein van Grouw and Mark Adams); Papua New Guinea National Museum and Art Gallery, Port Moresby, Papua New Guinea; Swedish Museum of Natural History, Stockholm, Sweden (Ulf Johansson) and the Natural history museum of Denmark, Copenhagen, Denmark (Peter Hosner). Lastly, we would like to thank Nikolay Oskolkov for assistance with bioinformatic analyses as part of the drop-in services and the Swedish Bioinformatics Advisory Program of the National Bioinformatics Infrastructure Sweden (NBIS).

## Author Contributions

**Conceptualization:** Ingo A. Müller, Mozes P. K. Blom, Knud A. Jønsson, Martin Irestedt.

**Data curation:** Ingo A. Müller, Samyuktha Rajan, Mozes P. K. Blom, Knud A. Jønsson, Martin Irestedt.

**Formal analysis:** Ingo A. Müller, Filip Thörn, Samyuktha Rajan, Per G. P. Ericson.

**Funding acquisition:** Ingo A. Müller, Knud A. Jønsson, Martin Irestedt.

**Investigation:** Ingo A. Müller, Filip Thörn, Samyuktha Rajan, Per G. P. Ericson, John P. Dumbacher.

**Methodology:** Ingo A. Müller, Filip Thörn, Samyuktha Rajan, Per G. P. Ericson, Mozes P. K. Blom, Martin Irestedt.

**Project administration:** Mozes P. K. Blom, Knud A. Jønsson, Martin Irestedt.

**Resources:** Ingo A. Müller, Per G. P. Ericson, John P. Dumbacher, Gibson Maiah, Mozes P. K. Blom, Knud A. Jønsson, Martin Irestedt.

**Software:** Ingo A. Müller, Filip Thörn, Samyuktha Rajan, Per G. P. Ericson, Mozes P. K. Blom.

**Supervision:** Mozes P. K. Blom, Knud A. Jønsson, Martin Irestedt.

**Validation:** Ingo A. Müller, Per G. P. Ericson, Mozes P. K. Blom, Knud A. Jønsson, Martin Irestedt.

**Visualization:** Ingo A. Müller, Filip Thörn, Samyuktha Rajan, Per G. P. Ericson.

**Writing – original draft:** Ingo A. Müller, Knud A. Jønsson, Martin Irestedt.

**Writing – review & editing:** Ingo A. Müller, Filip Thörn, Samyuktha Rajan, Per G. P. Ericson, John P. Dumbacher, Gibson Maiah, Mozes P. K. Blom, Knud A. Jønsson, Martin Irestedt.

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
