## [Decision Letter · Decision Letter 0]

11 Dec 2023

PONE-D-23-33496Species-specific dynamics may cause deviations from general biogeographical predictions – evidence from a population genomics study of a New Guinean endemic passerine bird family (Melampittidae).PLOS ONE

Dear Dr. Müller,

Thank you for submitting your manuscript to PLOS ONE. After careful consideration, we feel that it has merit but does not fully meet PLOS ONE’s publication criteria as it currently stands. Therefore, we invite you to submit a revised version of the manuscript that addresses the points raised during the review process.

Both reviewers think that the study is a good contribution to an understudied taxon and that the study provides valuable data. However, they also raise some concerns about the analyses and make some suggestions on how to improve the manuscript. 

We look forward to receiving your revised manuscript.

Kind regards,

Sven Winter

Academic Editor

PLOS ONE

Journal Requirements:

2. To comply with PLOS ONE submissions requirements, please provide the following information in the Methods section of the manuscript and in the “Ethics Statement” field of the submission form (via “Edit Submission”):  

*  Please indicate whether an animal research ethics committee prospectively approved this research or granted a formal waiver of ethics approval.*  Please enter the name of your Institutional Animal Care and Use Committee (IACUC) or other relevant ethics board. Also include an approval number if one was obtained.

*   If anesthesia, euthanasia, or any kind of animal sacrifice is part of the study, please include briefly in your statement which substances and/or methods were applied.

For additional information about PLOS ONE submissions requirements for ethics oversight of animal work, please refer to http://journals.plos.org/plosone/s/submission-guidelines#loc-animal-research  

5. We note that [Figure 1] in your submission contain [map/satellite] images which may be copyrighted. All PLOS content is published under the Creative Commons Attribution License (CC BY 4.0), which means that the manuscript, images, and Supporting Information files will be freely available online, and any third party is permitted to access, download, copy, distribute, and use these materials in any way, even commercially, with proper attribution. For these reasons, we cannot publish previously copyrighted maps or satellite images created using proprietary data, such as Google software (Google Maps, Street View, and Earth). For more information, see our copyright guidelines: http://journals.plos.org/plosone/s/licenses-and-copyright.

Reviewers' comments:

Reviewer's Responses to Questions

**Comments to the Author**

1. Is the manuscript technically sound, and do the data support the conclusions?

Reviewer #1: Partly

Reviewer #2: Yes

2. Has the statistical analysis been performed appropriately and rigorously? 

Reviewer #1: Yes

Reviewer #2: Yes

3. Have the authors made all data underlying the findings in their manuscript fully available?

Reviewer #1: Yes

Reviewer #2: Yes

4. Is the manuscript presented in an intelligible fashion and written in standard English?

Reviewer #1: Yes

Reviewer #2: Yes

5. Review Comments to the Author

Reviewer #1: Interesting study, in which the authors seem to have managed to produce good quality data from museum samples. However, in terms of the analyses, and interpretation and presentation of results, the study/paper needs in my opinion substantial improvement. See my comments below. The major comments are indicated with 'MAJOR'.

Abstract:

MAJOR: 50% of the abstract is intro, 30% methods, with effectively only one sentence describing the results. If in the conclusion you claim that certain populations should be raised to species level, this should be mentioned in the abstract. Also, refer back to the title: which species-specific dynamics? And how do the study species deviate?

Introduction

Line 68-70: what would young lineages be better adapted to lowland habitats?

Line 85: why ‘in accordance’? Why do you assume that the Lesser M. is an old lineage?

Line 91-93: Are mid-elevations to be considered here as ‘lowland’?

Line 95-98: specify here number of individuals sequenced, and to which depth. Also specify roughly which kind of analyses.

Line 96: MAJOR: while genomic data allows indeed to determine present-day population structure with the methods employed in this study (pca, phylogeny, admixture), but more methods are needed to determine how ‘habitat connectivity across space and time has shaped differentiation’

Line 99-106: Wouldn’t you expect a priori the opposite? Namely, the high-elevation species (Lesser. M) to have low connectivity between mountain-tops and hence to have population structure?

Methods

Line 109-117: why so few modern samples? Are samples difficult to acquire?

Line 126: 2x100bp read length is deliberate?

Line 127-128: How much data per sample? Expected genome size, targeted depth?

Line 129-138: While I appreciate the efforts made to make the work reproducible using a pipeline, the authors still need to describe here the settings of each software used by the pipeline (e.g., samtools and bwa). For instance, which base quality, mapping quality, and alignment score thresholds were used? Furthermore, all software used by the pipeline, needs to be mentioned here, and referenced.

Line 142: Because the authors so far did not mention the mean sequencing depth, it is hard to evaluate whether these thresholds are appropriate. The mitogenome depth is usually much higher than the nuclear depth, and hence 20x could be low. Could the authors present a summary of mean coverage of both the nuclear and mito genome?

Line 152: Mitochondrial genomes (!) might be effectively haploid, but in reality they are highly polyploid: each cell contains hundreds or thousands of copies which are independently proliferating, which may also cause heterozygous calls.

Line 155: for museum samples, a depth below 4x is not too bad. Why freebayes? Other genotyping software (like bcftools, GATK or ANGSD) have been reported to have higher accuracy.

Line 160-163: Again, it does not suffice to only mention the pipeline. Even though this information could be looked up at the Github page, it needs to be specified here which settings were used to run IQtree2 and ASTRAL.

Line 162: Running IQtree (maximum likelihood approach) on an input dataset of SNPs (diploid, recombining loci), violates underlying assumptions of ML tree building.

Line 162: Better to use here the word ‘locus tree’ rather than ‘gene tree’

Line 165: MAJOR: which of the below mentioned statistics estimate levels of differentiation? In reality, all statistics (nucleotide diversity, heterozygosity and SFS) estimate genetic diversity (with Tajima’s D testing for neutrality). PCA and admixture show population structure, but do not estimate the level of population differentiation.

Line 170: MAJOR: With only 6 individuals per population, does it really make sense to try to reconstruct the SFS?

Line 173: Admixture analyses were run…

Line 178: MAJOR: Does it really make sense to calculate Tajima’s D for an entire chromosome? This test is designed for a single, non-recombining locus.

Results

Line 232-247: move to methods section.

Line 295-317: a population split of 4-5 Mya, would this not imply these populations are in fact different species? Else, is it possible that the PSMC settings inflate the divergence time estimate?

Discussion

Line 340: how do we know it is an old lineage? When does a lineage quality as ‘old’?

Line 349: MAJOR: a population split of 4-5 Mya, would this not imply these populations are in fact different species? Else, is it possible that the PSMC settings inflate the divergence time estimate?

Line 368: See my previous comment for line 99-106: Wouldn’t you expect a priori the opposite? Namely, the high-elevation species (Lesser. M) to have low connectivity between mountain-tops and hence to have population structure?

Conclusion:

Line 396: Which intrinsic properties?

Line 399-401: MAJOR: Perhaps I overread, but how did you estimate levels of divergence, other than using hPSMC to estimate divergence times? As mentioned in a previous comment, a split time of 4-5 Mya would indeed suggest species status. But to be better able to evaluate this claim, it would be useful if the authors would estimate levels of genetic divergence between the populations (i.e. Dxy).

Figures

Figure 1. MAJOR colour coding of eastern populations does not correspond between A and B. Add to the figure legend (following the species names) the elevation ranges of the two species. Throughout the text, the authors could make clearer (or reminder the reader) that the Lesser M. occurs at high elevation, and the Greater M at mid elevation. In the caption of figure 1, specify which software/method has been used to construct the phylogeny.

Figure 2. MAJOR These analyses only qualitatively assess population structure, but do not provide quantitative estimates (e.g. Dxy or Fst). The legend (e.g. Vogelkop, Weyland, etc) does not correspond with the labels of the admixture plot. Regarding the admixture plot: I find it suspicious that there is no evidence for admixture whatsoever. Especially between the central populations, which are geographically close to each other, should there not be any exchange? Regarding the He-plot, could you present He per population (e.g., west, east, central) rather than per species?

Figure 3. MAJOR Since PSMC is used to evaluate split times, why not plot the curves on top of each so that it is better possible to evaluate when the lines diverge? It should also be made more clear which populations/species each tile represents (the sample names are not informative). Where are the bootstraps? In figure 3c, one of the samples need to be corrected, in order to overlap (which they clearly do, just need to be corrected for difference in depth). Where is the hPSMC plot?

Reviewer #2: This study provides a novel genetic analysis of a so-far poorly-studied passerine bird family (Melampittidae). It is a huge merit of the team that they managed to analyse all seven known museum specimens of Megalampitta gigantaea. Although this species is currently not considered to be under threat (least concern, IUCN), there is a considerable knowledge gap on that species’ distribution and intraspecific diversification (e.g. whether it comprises more than one evolutionary significantly unit). This emphasizes the relevance of museomics for current biodiversity research. The research team is known for their strong expertise of collection-based genetics/genomics and the use (and development) of specific protocols for historic DNA analysis. The results shade new light on the cryptic diversification in this group, e.g. despite being considered a monotypic species M. lugubris turned out to comprise several distinct genetic lineages restricted to different mountain ranges on New Guinea. This is a fine study that is highly recommendable for publication. In the following I am commenting on a few aspects of the study that should be outlined or explained in a little more detail in a revised manuscript.

l. 63: “older taxa are often found at higher elevations, while young lineages that are generally widespread, good dispersers and show little differentiation inhabit the lowlands”. Following a general introduction on mountain biodiversity and evolutionary patters in mountain specialists, this reads as if it was a generalized statement. However, this seems to be a rather characteristic pattern on New Guinea (see l. 71), “island systems” (l. 65) or in other tropical ecosystems, whereas in other big mountain systems of the Earth, this does not seem to be the common pattern. For the Himalayas (and the Andes) see Fjieldsa et al. (2012: Fig. 1) with both mountain systems showing highest richness of both youngest and oldest species. Furthermore, for the Himalayas Price et al. (2014) stated that „we found that the average age of separation of species in the assemblage declines monotonically with elevation, rather than being lowest in the most species-rich elevational band”. I think in this introductory paragraph it is important to emphasize that it is not the general pattern for any mountain system that evolutionary ancient species occur on top of the mountain.

l. 82: “recent genetic results have placed the family as sister to crows (Corvidae) and shrikes (Laniidae) with an estimated divergence time from these at ca. 17 Mya [31]”

This statement refers to Oliveros et al. (2019), however, this year a new study has been published by

McCollough et al. 2023: Ornithology, 2023, 140, 1–11 https://doi.org/10.1093/ornithology/ukad025. Their phylogeny based on UCEs placed Eurocephalidae as the sister to Corvidae, Platylophidae sister to Laniidae and Melampittidae was sister to all four. For completeness, this study should be acknowledged here.

l. 115: “the work is mainly based on old museum specimens for which the Nagoya Protocol does not apply.” I do not expect a reply from the authors to this comment, because this is just a friendly, but to my feeling important reminder. I would like to send out a warning to the authors rather not to use such vague statements. The word “mainly” implies that “not all” samples used are old and could evoke the impression of decision makers (including the persons in charge on the NFPs of a country of origin), that for some of the material the Nagoya Protocol might theoretically apply and that this must have been checked before publication (even before performing any genetic work on the material). Moreover, I think this statement is unnecessary, because at least to my information Papua New Guinea (PNG) is not a signatory country of the Nagoya Protocol (check here)

https://www.cbd.int/abs/nagoya-protocol/signatories/

whereas, nevertheless PNG has established an NFP for the Nagoya Protocol. So, this is one of the many very tricky cases and potential pitfalls for us scientists (check here)

https://www.cbd.int/countries/?country=pg

I assume that even the two “fresh tissue samples” (l. 111) had been assessed before October 2014; if so, then I would suggest either stating that Nagoya does not apply to the study material, because all samples were assessed before that date – however, in that case, that remark is rather unnecessary and I would recommend rather deleting this statement, if journal policies do not require adding a disclaimer on compliance with the Nagoya Protocol.

l. 117: “permits are available”; it would be good to cite these (including permit numbers) in the acknowledgements (collecting or research permits in the country of origin etc.).

l. 207: “We applied the simple 2% rule…” Indeed, this is a very simplistic approach, and I am not sure whether I correctly understand, how this was actually done. I think we can infer from the reference to Weir & Schluter (2008) that the empirical cytochrome-b rate was apparently applied across the entire mitogenome. To me it seems that pairwise distances (inferred from whole mitogenomes including tRNAs, rRNAs and non-coding regions like the D-loop) among taxa/clades simply were transferred into split ages using the cytb rate (the statement in l. 205/206 evokes this impression). Considering that mitochondrial genes evolve a different substitution rates (in fact quite different between coding and non-coding regions; ; compare Lerner et al. 2011: Curr Biol 21, 1838-1844), this is not a very precise approach (and I would not consider this the state of the art). The more convincing method would be reconstructing a time-calibrated mitochondrial phylogeny, using for example the thirteen coding-markers of the mitogenome (as thirteen separate partitions in BEAUti) and applying empirical rates to each partition (e.g. from Lerner et al. 2011). If the simple approach is preferred, then it should be limited to the cytb fragments of the Melampittidae mitogenomes, because the Weir & Schluter estimate refers only to cytb.

l. 224: “Elements were measured as continuous sound traces and then grouped into syllables within each vocalisation (each vocalisation contained only one syllable).”

It is unclear to me what was done here. From that scarce information I think it is not possible to infer, how song characters were quantified, i.e. which sound parameters (frequency, time) were actually measured. Maybe this information is hidden in the statement on the DTW algorithm and the link to reference 62 (l. 226/227). Nevertheless, this vague information is not helpful to understand the differences in songs among the two species (except that one is more variable than the other). If 10 principal components were extracted these should normally correlate with song parameters (to be inferred from factor loads). So, as the reader I would like to know: What does differentiation along PC1 actually tell us? (Fig. 4; ~ 44% of variation) Which acoustic traits correlate with PC1? Are these differences in pitch, speed, complexity? Given that many acoustic features change with habitat density that might also relate to the two study species (mountain forests versus [more open?] karst habitats), this seems relevant information to me. This should be outlined in much more detail. A figure showing sonagrams would be actually helpful, because from the text I have no idea of the specific vocalizations (traits measured could then also be shown in one of the sonagrams). A closer check of sonagrams, could also be helpful for interpretation of the “significant outlier” in M. lugubris (legend of fig. 4; page 14). What means an “odd vocalization” in this context? Given the apparently high uncertainty of the context of a recorded vocalization (as outlined in Acoustic recordings and analysis), this one vocalization might not even be homologous to the other vocalizations of M. lugubris. Considering that even the distinction between “calls” and “songs” was unclear in this study, the vocalizations in general should be described and illustrated in much more detail.

l. 324: “greater acoustic diversity” Indeed, that can be inferred from the scatterplot, but this is not the same as a greater “vocal differentiation” (l. 367/68). The latter would imply diversified vocal groups, e.g. among mountain systems. At least, in the context of that paragraph it comes across that way, because the previous sentence directly refers to differences among the two species in “populations structure” (M. gigantea does not show a clear [genetic] population structure in contrast to M. lugubris; l. 364). The next sentence says that the vocal pattern is similar, however, although clear genetic clusters have been shown for M. lugburis, this is not true for the songs. Only because the songs are more variable, this does not mean, that this greater variation corresponds to different acoustic entities (dialects for example). At least this has not been shown for songs in the analysis.

l. 269: “We recover the same pattern of lower levels of differentiation in M. gigantaea compared to M. lugubris in the PCAs (Fig. 2 A) …” I do not understand this statement. Isn’t the whole paper on the clear differences between the species, e.g. M gigantaea being one very homogenic cluster in the PCA, and M. lugubris being represented by two clear PCA clusters and even a clear clustering structure for k=6. How does this conform with the statement in l. 281 that “Both heterozygosity and nucleotide diversity were significantly lower in M. gigantaea than in M. lugubris”? How can that paragraph be started with the statement that patterns of low (intraspecific) differentiation were the same in the two study species?

Minor

In legend of figures 1, 2, 3 4 attention should be paid to the correct italic format of all scientific names.

6. PLOS authors have the option to publish the peer review history of their article (what does this mean?). If published, this will include your full peer review and any attached files.

Reviewer #1: No

Reviewer #2: No

---

## [Author Response · Author response to Decision Letter 0]

26 Jan 2024

Journal Requirements:

RESPONSE: We have gone over the manuscript and made changes where necessary to fulfill your formatting requirements.

2. To comply with PLOS ONE submissions requirements, please provide the following information in the Methods section of the manuscript and in the “Ethics Statement” field of the submission form (via “Edit Submission”)

RESPONSE: We have updated our methods section and ethics statement to clarify that our research has been performed on samples stored in museum collections for which no ethical approval is required. Relevant collection and export permits were also added.

+

RESPONSE: We will make sure our sequences will be uploaded onto the European Nucleotide Archive (ENA) which is synchronised with NCBI until a potential acceptance of the manuscript and will provide you with the relevant accession numbers once available.

5. We note that [Figure 1] in your submission contain [map/satellite] images which may be copyrighted.

RESPONSE: Map and satellite images for this figure were obtained from the United States Geological Survey (USGS, open domain) and geoBoundaries (https://www.geoboundaries.org/), an open database built by the geoLab at the College of William & Mary, which makes all data available under a CC BY 4.0 license. Nonetheless, we have attached a signed permission form by the geoLab confirming their permission to use this data.

Reviewer #1: Interesting study, in which the authors seem to have managed to produce good quality data from museum samples. However, in terms of the analyses, and interpretation and presentation of results, the study/paper needs in my opinion substantial improvement. See my comments below. The major comments are indicated with 'MAJOR'.

Abstract:

MAJOR: 50% of the abstract is intro, 30% methods, with effectively only one sentence describing the results. If in the conclusion you claim that certain populations should be raised to species level, this should be mentioned in the abstract. Also, refer back to the title: which species-specific dynamics? And how do the study species deviate?

RESPONSE: We have revised the abstract to accommodate this critique. Specifically, we have shortened the intro and added information about the potential elevation of Lesser Melampitta populations to full species. We also refer back to the title. 

Introduction

Line 68-70: what would young lineages be better adapted to lowland habitats?

RESPONSE: This sentence summarises the concept of taxon cycles in which taxa go through phases of expansions (lowland generalists) and contractions (highland specialisation). The idea is that from time-to-time lowland generalists colonise new islands. As they are generalists, they tend to colonise peripheral (coastal) habitat. Over time these taxa move inland and upwards and become specialists. Little is known about the order of this. In any case, when it comes to taxon cycles, the new colonisers are thought to be “pioneer species” that can exploit the unstable coastal habitats better than more specialised old species. We have tried to summarise this very briefly in the introduction and have now added the word “generalist” to the text to indicate that the new lowland colonisers are expected to be generalists. 

Line 85: why ‘in accordance’? Why do you assume that the Lesser M. is an old lineage?

RESPONSE: The family Melampittidae with only the two species investigated herein represent a very deep linage within Corvides and is one of a few very distinct and old lineages (16.1 My) in New Guinea. Moreover, the two currently recognised species within this family are placed in different genera and diverged from each other about 10 Mya. Although old is a relative term, these two genera represent old linages within Corvides and the divergence between these two species/genera is also old compared to other sister taxa divergences in this region. 

Line 91-93: Are mid-elevations to be considered here as ‘lowland’?

RESPONSE: We apologise that this was not clear from the text, but we do not consider the habitat of M. gigantaea as lowlands, but neither as high elevations. We wanted to point out that this species has moved out of the coastal lowlands to some extent, but not into the high elevations that we normally observe for old/relictual taxa (as frequently observed in other avian groups on New Guinea). We consider here a scenario where rather than moving even higher up, M. gigantaea moved into suboptimal karst habitats (at mid-elevations) where they would also encounter less competition.

Line 95-98: specify here number of individuals sequenced, and to which depth. Also specify roughly which kind of analyses.

RESPONSE: This is a valid point and we have added this information to the introduction. More detailed numbers can be found at the beginning of the results and in supplementary table S1.

Line 96: MAJOR: while genomic data allows indeed to determine present-day population structure with the methods employed in this study (pca, phylogeny, admixture), but more methods are needed to determine how ‘habitat connectivity across space and time has shaped differentiation’

RESPONSE: Thank you for this important note. We have deleted and rephrased this segment to be more precise in the aim of this study and which questions can be answered with the applied methodology.

Line 99-106: Wouldn’t you expect a priori the opposite? Namely, the high-elevation species (Lesser. M) to have low connectivity between mountain-tops and hence to have population structure?

RESPONSE: Intuitively, we agree that we would expect a high-elevation species to be more isolated across a mountain range than a species inhabiting lower elevations. However, in this study - due to the large distances between the few distributional data points - M. gigantaea was expected to exhibit a stronger structure between populations. Additionally, since M. gigantaea is specialised on karst environments, we would expect the species to struggle outside these environments and be strongly isolated.

Methods

Line 109-117: why so few modern samples? Are samples difficult to acquire?

RESPONSE: Yes, the low number of modern samples is due to the difficulty of collection. For example, co-author Knud Jønsson spent 6 months in the field in Papua New Guinea within the distribution of M. lugubris and only ever collected two M. lugubris samples. Additionally, sampling in Indonesia has become almost impossible due to government regulations. For M. gigantaea sampling is even more difficult as we were able to obtain the only fresh sample collected within the last 50 years as well as all 6 historical specimens that are available in museum collections worldwide. 

Line 126: 2x100bp read length is deliberate?

RESPONSE: We choose a read length of 2x100 bp in our museomics projects as our fragments rarely reach sizes longer than 200 bp (the average length of our fragments ranges from about 90 bp to 130 bp, see e.g. Irestedt et al. (2022)) that would make it worthwhile to sequence with the 2x150 bp chemistry. 

Irestedt, M., Thörn, F., Müller, I. A., Jønsson, K. A., Ericson, P. G., & Blom, M. P. (2022). A guide to avian museomics: Insights gained from resequencing hundreds of avian study skins. Molecular Ecology Resources, 22(7), 2672-2684.

Line 127-128: How much data per sample? Expected genome size, targeted depth?

RESPONSE: We have added the requested information to the text. Our expected genome size was about 1 GB, given that the genome sizes in passerine birds is conserved and by the size of our chosen reference genome (Corvus cornix, genome size: 1.031 GB). As we did not sequence any de novo genome of Melampittidae we do not have a more precise estimate.

Line 129-138: While I appreciate the efforts made to make the work reproducible using a pipeline, the authors still need to describe here the settings of each software used by the pipeline (e.g., samtools and bwa). For instance, which base quality, mapping quality, and alignment score thresholds were used? Furthermore, all software used by the pipeline, needs to be mentioned here, and referenced.

RESPONSE: Thank you for the comment. We have added each tool and its version of all tools that are part of the Nextflow workflows. All implemented flags are now included in supplementary file S2. Bwa-mem2 and samtools were both run with default settings as we perform the relevant filtering steps in subsequent downstream analyses anyway.

Line 142: Because the authors so far did not mention the mean sequencing depth, it is hard to evaluate whether these thresholds are appropriate. The mitogenome depth is usually much higher than the nuclear depth, and hence 20x could be low. Could the authors present a summary of mean coverage of both the nuclear and mito genome?

RESPONSE: We have expanded supplementary table S1 to distinguish depth-of-coverage (DoC) of the nuclear genome and the mitochondria. The median DoC 786 x of the mitochondria is of course much higher than in the nuclear genome, a minimum DoC of 20 x is in our opinion still sufficient to make confident variant calls with low biases from erroneous reads.

Line 152: Mitochondrial genomes (!) might be effectively haploid, but in reality they are highly polyploid: each cell contains hundreds or thousands of copies which are independently proliferating, which may also cause heterozygous calls.

RESPONSE: This is absolutely true and a valid point. While we expect a certain number of heterozygous calls due to heteroplasmy, we saw a significantly higher number of such sites in six individuals (> 50 sites although most of these had more than 100-200 heterozygous sites, compared to 0-5 sites in most other individuals) and these sites were always concentrated in the same region across these individuals. As the flanking regions of the reads that produce these blocks were not of mitochondrial origin, we consider it highly likely that they represent NUMTs and has thus been deleted. We would not expect such a concentrated pattern due to random mutations in independent mitochondria especially not if only occurred in some individuals. 

Line 155: for museum samples, a depth below 4x is not too bad. Why freebayes? Other genotyping software (like bcftools, GATK or ANGSD) have been reported to have higher accuracy.

RESPONSE: A depth of coverage around 4 x is indeed not too bad, but it’s also not great in comparison to coverages typically seen for sequencing data of fresh tissue samples. Genotyping software such as GATK was developed for analysing data from model organisms where we often have a good reference point to calibrate our variant calling. In other words, we can optimise our parameter settings for variant call recalibration using truth and training sets. However, this is much more difficult to do for non-model organisms and for historical datasets where there is a higher error rate. At the same time, 4 x depth of coverage is indeed much higher than typically seen for ancient DNA data where genotype likelihood methods such as employed by ANGSD excel. We therefore opted for a holistic approach using both ANGSD (for admixture analyses, PCAs and estimates of π, Tajima’s D, Heterozygosity) as well as freebayes, because the latter allows the use of a population prior when variant calling. This population information is used for the imputation of genotypes across a population and should therefore limit potential residual deamination patterns in historical samples. We are unaware of any reports that have demonstrated a higher accuracy for other genotyping software with this kind of datasets.

Line 160-163: Again, it does not suffice to only mention the pipeline. Even though this information could be looked up at the Github page, it needs to be specified here which settings were used to run IQtree2 and ASTRAL.

RESPONSE: Thank you, we have now expanded this section to mention all major steps and tools citing the respective creators. We have also modified supplementary file S2 to include all implemented flags and filters. 

Line 162: Running IQtree (maximum likelihood approach) on an input dataset of SNPs (diploid, recombining loci), violates underlying assumptions of ML tree building.

RESPONSE: We don’t think that a concatenated dataset violates the underlying assumptions of ML tree building itself. In fact, this is a very common approach to infer the prevailing phylogenetic signal across a (whole-genome) dataset (see e.g. Lopes, F., et al. Systematic Biology (2021), Bein, B., et al. Marine Biology (2023)). However, we do agree that it does not account for any possible variation in coalescent histories between loci across and between chromosomes (i.e. where there has been a recombination event). We therefore complement the ML inference of concatenated data with a summary-coalescent species tree estimation based approach (ASTRAL3) which is based on genomic windows of different lengths (here we ran 2000, 5000, 10000 and 20000 bp windows) rather than concatenation.

Lopes, F., Oliveira, L. R., Kessler, A., Beux, Y., Crespo, E., Cárdenas-Alayza, S., ... & Bonatto, S. L. (2021). Phylogenomic discordance in the eared seals is best explained by incomplete lineage sorting following explosive radiation in the southern hemisphere. Systematic biology, 70(4), 786-802.

Bein, B., Lima, F. D., Lazzarotto, H., Rocha, L. A., Leite, T. S., Lima, S. M., & Pereira, R. J. (2023). Population genomics of an Octopus species identify oceanographic barriers and inbreeding patterns. Marine Biology, 170(12), 161.

Line 162: Better to use here the word ‘locus tree’ rather than ‘gene tree’

RESPONSE: You are correct, since we are doing a window-based the term gene tree is not correct here. In the rephrased segment we have avoided this term. 

Line 165: MAJOR: which of the below mentioned statistics estimate levels of differentiation? In reality, all statistics (nucleotide diversity, heterozygosity and SFS) estimate genetic diversity (with Tajima’s D testing for neutrality). PCA and admixture show population structure, but do not estimate the level of population differentiation.

RESPONSE: Thank you very much for pointing out this issue. We realise that the term differentiation was not used correctly here as we haven’t employed a direct measure of it. We have corrected all mentions of differentiation to either refer to population structure or genetic diversity.

Line 170: MAJOR: With only 6 individuals per population, does it really make sense to try to reconstruct the SFS?

RESPONSE: While the sample size is close to the recommended minimum for population level estimates, we still believe that our sampling scheme allows for a decent estimate of our calculated statistics as we included all available samples of M. gigantaea which cover its entire distribution across New Guinea as well as M. lugubris individuals that all cover different localities within the distribution of its populations.

Line 173: Admixture analyses were run…

RESPONSE: Thank you for pointing out this grammatical error, we have now corrected it in the text.

Line 178: MAJOR: Does it 

---

## [Decision Letter · Decision Letter 1]

23 Feb 2024

PONE-D-23-33496R1Species-specific dynamics may cause deviations from general biogeographical predictions – evidence from a population genomics study of a New Guinean endemic passerine bird family (Melampittidae).PLOS ONE

Dear Dr. Müller,

Thank you for submitting your manuscript to PLOS ONE. After careful consideration, we feel that it has merit but does not fully meet PLOS ONE’s publication criteria as it currently stands. Therefore, we invite you to submit a revised version of the manuscript that addresses the points raised during the review process.

 The last revision greatly improved the manuscript. Reviewer #1 has a few more minor requests that should be easily implemented before acceptance. 

We look forward to receiving your revised manuscript.

Kind regards,

Sven Winter

Academic Editor

PLOS ONE

Journal Requirements:

Reviewers' comments:

Reviewer's Responses to Questions

**Comments to the Author**

1. If the authors have adequately addressed your comments raised in a previous round of review and you feel that this manuscript is now acceptable for publication, you may indicate that here to bypass the “Comments to the Author” section, enter your conflict of interest statement in the “Confidential to Editor” section, and submit your "Accept" recommendation.

Reviewer #1: All comments have been addressed

2. Is the manuscript technically sound, and do the data support the conclusions?

Reviewer #1: Yes

3. Has the statistical analysis been performed appropriately and rigorously? 

Reviewer #1: Yes

4. Have the authors made all data underlying the findings in their manuscript fully available?

Reviewer #1: Yes

5. Is the manuscript presented in an intelligible fashion and written in standard English?

Reviewer #1: Yes

6. Review Comments to the Author

Reviewer #1: I am satisfied with the sensibly replies to my comments, as well as the corrections made to the manuscript. I have just one remaining request/suggestion, namely to estimate the level of divergence between lineages using the Dxy-estimate. This is crucial to assess the species status. The authors have argued that sample size prohibits this calculation, but actually Dxy is insensitive to sample size (see also one of my replies below). For reference, see also: Roux et al. 2016 Shedding Light on the Grey Zone of Speciation along a Continuum of Genomic Divergence

A few replies (referring to line numbers of comments of first revision round):

Line 109-117: The authors could consider adding this information to the methods section, so the reader better appreciate the efforts that the authors have put in acquiring this dataset.

Line 126: same: useful information, why not include in methods section?

Line 142: Agreed, 20x is sufficient, but on the other hand, which factors could cause a site or region to have a depth of 20 if the mean is 786? It makes such regions look ‘suspicuous’.

Line 152: Thanks for the clarification, which even gives me a potential explanation of patterns observed in my own datasets.

Line 155: Again, a concise summary of these considerations could be added to the methods section, in order for the reader to appreciate the underlying rationale.

Line 162: I am not fully convinced by this answer. Common approaches are not necessarily correct approaches. ML-phylogenetic inferences aims to reconstruct the most likely phylogenetic model, including the parameters u (mutation rate) and t (branch length), given the data. For a concatenated set of independently evolving loci, each with its own u and t, would this mean that in practice the method aims to infer the most likely average values of u and t (?). And if the input dataset is unphased, how does the method deal with ambiguous sites stemming from heterozygosity?

Line 178: Actually, I have to correct here myself. In the original paper, Tajima applies the test both to single-locus and multi-locus datasets.

Line 295-317: As a ‘second-opinion’, the authors could calculate Dxy (mean absolute genetic distance), for example using the software PIXY, or using the python scripts from the github page of Simon Martin (distMat.py, or popgenWindows.py –analysis popPairDist). Note that when inputting snp data, you would afterwards have to correct for this by multiplying the output estimates by the proportion of variable sites.

Line 399-401: Unlike Fst, Dxy is not sensitive to sample size. This is one of the main advantages of Dxy over Fst. (Another advantage is that it is not effective by Ne.) Thus, Dxy is even valid in case each population is represented by a single individual only. When using entire genomes, single-locus stochastics are cancelled out by the law or large numbers.

7. PLOS authors have the option to publish the peer review history of their article (what does this mean?). If published, this will include your full peer review and any attached files.

Reviewer #1: No

---

## [Author Response · Author response to Decision Letter 1]

22 Mar 2024

Journal Requirements: Please review your reference list to ensure that it is complete and correct. If you have cited papers that have been retracted, please include the rationale for doing so in the manuscript text, or remove these references and replace them with relevant current references. Any changes to the reference list should be mentioned in the rebuttal letter that accompanies your revised manuscript. If you need to cite a retracted article, indicate the article’s retracted status in the References list and also include a citation and full reference for the retraction notice.

RESPONSE: We have double checked our reference list for completion and we did not see any citation that has been retracted at this point. Based on the reviewer’s suggestions we have additionally included the following references that were not used in the original manuscript (no reference has been removed):

32. McCullough JM, Hruska JP, Oliveros CH, Moyle RG, Andersen MJ. Ultraconserved elements support the elevation of a new avian family, Eurocephalidae, the white-crowned shrikes. Ornithology. 2023 Jul 11;140(3):ukad025.

42. Andrews S. Babraham Bioinformatics - FastQC A Quality Control tool for High Throughput Sequence Data [Internet]. 2010. Available from: https://www.bioinformatics.babraham.ac.uk/projects/fastqc/

43. s4hts/HTStream [Internet]. Software (for) High Throughput Sequencing; 2023 [cited 2024 Jan 15]. Available from: https://github.com/s4hts/HTStream

44. Bolger AM, Lohse M, Usadel B. Trimmomatic: a flexible trimmer for Illumina sequence data. Bioinformatics. 2014 Aug 1;30(15):2114–20. 

45. Zhang J, Kobert K, Flouri T, Stamatakis A. PEAR: a fast and accurate Illumina Paired-End reAd mergeR. Bioinformatics. 2014 Mar 1;30(5):614–20. 

46. Shen W, Le S, Li Y, Hu F. SeqKit: A Cross-Platform and Ultrafast Toolkit for FASTA/Q File Manipulation. PLOS ONE. 2016 Oct 5;11(10):e0163962.

49. Danecek P, Bonfield JK, Liddle J, Marshall J, Ohan V, Pollard MO, et al. Twelve years of SAMtools and BCFtools. GigaScience [Internet]. 2021 Feb;10(2). Available from: https://doi.org/10.1093/gigascience/giab008

50. Okonechnikov K, Conesa A, García-Alcalde F. Qualimap 2: advanced multi-sample quality control for high-throughput sequencing data. Bioinformatics. 2016 Jan 15;

32(2):292–4. 

51. Neukamm J, Peltzer A, Nieselt K. DamageProfiler: fast damage pattern calculation for ancient DNA. Bioinformatics. 2021 Oct 25;37(20):3652–3.

54. Hahn C, Bachmann L, Chevreux B. Reconstructing mitochondrial genomes directly from genomic next-generation sequencing reads—a baiting and iterative mapping approach. Nucleic Acids Res. 2013 Jul 1;41(13):e129.

58. Stegemiller MR, Redden RR, Notter DR, Taylor T, Taylor JB, Cockett NE, et al. Using whole genome sequence to compare variant callers and breed differences of US sheep. Front Genet [Internet]. 2023 Jan 4 [cited 2024 Mar 22];13. Available from: https://www.frontiersin.org/journals/genetics/articles/10.3389/fgene.2022.1060882/full

60. Nguyen LT, Schmidt HA, von Haeseler A, Minh BQ. IQ-TREE: A Fast and Effective Stochastic Algorithm for Estimating Maximum-Likelihood Phylogenies. Mol Biol Evol. 2015 Jan 1;32(1):268–74. 

61. Zhang C, Rabiee M, Sayyari E, Mirarab S. ASTRAL-III: polynomial time species tree reconstruction from partially resolved gene trees. BMC Bioinformatics. 2018 May 8;19(6):153. 

62. Rabiee M, Sayyari E, Mirarab S. Multi-allele species reconstruction using ASTRAL. Mol Phylogenet Evol. 2019 Jan 1;130:286–96.

74. Lerner HRL, Meyer M, James HF, Hofreiter M, Fleischer RC. Multilocus Resolution of Phylogeny and Timescale in the Extant Adaptive Radiation of Hawaiian Honeycreepers. Curr Biol. 2011 Nov;21(21):1838–44.

Reviewer #1: I am satisfied with the sensibly replies to my comments, as well as the corrections made to the manuscript. I have just one remaining request/suggestion, namely to estimate the level of divergence between lineages using the Dxy-estimate. This is crucial to assess the species status. The authors have argued that sample size prohibits this calculation, but actually Dxy is insensitive to sample size (see also one of my replies below). For reference, see also: Roux et al. 2016 Shedding Light on the Grey Zone of Speciation along a Continuum of Genomic Divergence

RESPONSE: We hope we have addressed all of your suggestions and have now included estimates of Dxy. See below for our responses to each point.

A few replies (referring to line numbers of comments of first revision round):

Line 109-117: The authors could consider adding this information to the methods section, so the reader better appreciate the efforts that the authors have put in acquiring this dataset.

RESPONSE: We have added some additional information in the manuscript to explain our low number of modern samples.

Line 126: same: useful information, why not include in methods section?

RESPONSE: We now include a sentence in the methods in which we explain our choice of sequencing read length.

Line 142: Agreed, 20x is sufficient, but on the other hand, which factors could cause a site or region to have a depth of 20 if the mean is 786? It makes such regions look ‘suspicuous’.

RESPONSE: We agree that such a heavy dip from the average depth of coverage would look odd. Even in the mitochondria, we would expect certain regions to be more difficult to sequence and assemble, such as the control region, but also repetitive regions. Additionally, our chosen method takes a random subset of 5 000 000 reads, which is normally more than sufficient for a mitochondrial assembly, but due to its random nature there may still be a low chance that a certain region is covered by less reads within the subset.

Line 152: Thanks for the clarification, which even gives me a potential explanation of patterns observed in my own datasets.

RESPONSE: Happy to help, we have discussed this issue extensively in our group.

Line 155: Again, a concise summary of these considerations could be added to the methods section, in order for the reader to appreciate the underlying rationale.

RESPONSE: We have added a short justification to each of the applied variant calling methods.

Line 162: I am not fully convinced by this answer. Common approaches are not necessarily correct approaches. ML-phylogenetic inferences aims to reconstruct the most likely phylogenetic model, including the parameters u (mutation rate) and t (branch length), given the data. For a concatenated set of independently evolving loci, each with its own u and t, would this mean that in practice the method aims to infer the most likely average values of u and t (?). And if the input dataset is unphased, how does the method deal with ambiguous sites stemming from heterozygosity?

RESPONSE: The first step of the used workflow (nf-phylo) is to generate a consensus sequence for every chromosome/scaffold based on the vcf of each individual, with heterozygous and low/high coverage positions masked. These consensus sequences are then used as input for phylogenetic inference, not the SNP-data itself. Although we do not reach the point of parameters for each independent locus, we try to accommodate for this issue by inferring phylogenies at different windows sizes (between 2 000 – 20 000 bp) and verify that the topology remains consistent. Additionally, we compare topologies between concatenated and summary coalescent phylogenies to make sure that these are also similar. Another measure that gives us a sense of support from each locus is obtained through the site concordance factor which measures the proportion of sites within each window that support a certain branch.

Ambiguous heterozygous sites are not an issue in our approach as we remove all heterozygous sites during the filtering of our SNP data. We definitely lose some information from these sites by removing them, however it gives us more security due to the relatively low coverage of some of our samples. We have clarified this part in the text (previously it only said “masked”).

Line 178: Actually, I have to correct here myself. In the original paper, Tajima applies the test both to single-locus and multi-locus datasets.

RESPONSE: Thank you for the correction. Your original comment was still very much valid, and we have also discussed with other colleagues that it is unusual to present Tajima’s D as a single value for the entire genome as one may miss regions with a strong different signal otherwise. Although we did not intend on investigating the genomes on this scale within this study, it is still important to show changes in Tajima’s D across the entire genome.

Line 295-317: As a ‘second-opinion’, the authors could calculate Dxy (mean absolute genetic distance), for example using the software PIXY, or using the python scripts from the github page of Simon Martin (distMat.py, or popgenWindows.py –analysis popPairDist). Note that when inputting snp data, you would afterwards have to correct for this by multiplying the output estimates by the proportion of variable sites.

RESPONSE: We have included estimates of Dxy which are now part of the supplementary material (S6 Table). We chose to calculate it through a genotype likelihood-based approach so that we could include as many individuals as possible. The script is based on https://github.com/mfumagalli/ngsPopGen/blob/master/scripts/calcDxy.R and was only modified to perform multiple pairwise comparisons and provide a summary of global estimates. The main calculation of Dxy remains the same. The approach is described in the methods and codes are listed in S2 File.

Line 399-401: Unlike Fst, Dxy is not sensitive to sample size. This is one of the main advantages of Dxy over Fst. (Another advantage is that it is not effective by Ne.) Thus, Dxy is even valid in case each population is represented by a single individual only. When using entire genomes, single-locus stochastics are cancelled out by the law or large numbers.

RESPONSE: Thank you for the comment, although we were sceptical whether Dxy would really be representative for single individuals, we were pleasantly surprised to find that pairwise estimates were around the same range even when using different subpopulations (smaller or even consisting of a single individual) for the calculation.

---

## [Editor Report · Decision Letter 2]

27 Mar 2024

Species-specific dynamics may cause deviations from general biogeographical predictions – evidence from a population genomics study of a New Guinean endemic passerine bird family (Melampittidae).

PONE-D-23-33496R2

Dear Dr. Müller,

We’re pleased to inform you that your manuscript has been judged scientifically suitable for publication and will be formally accepted for publication once it meets all outstanding technical requirements.

Kind regards,

Sven Winter

Academic Editor

PLOS ONE